# Intrusion Detection in Internet of Things Systems: A Review on Design Approaches Leveraging Multi-Access Edge Computing, Machine Learning, and Datasets

**DOI:** 10.3390/s22103744

**Published:** 2022-05-14

**Authors:** Eric Gyamfi, Anca Jurcut

**Affiliations:** School of Computer Science, University College Dublin, D04 V1W8 Dublin, Ireland; anca.jurcut@ucd.ie

**Keywords:** Internet of Things, multi-access edge computing, network intrusion detection system, machine learning, anomaly systems, IoT device, offloading, distributed NIDS

## Abstract

The explosive growth of the Internet of Things (IoT) applications has imposed a dramatic increase of network data and placed a high computation complexity across various connected devices. The IoT devices capture valuable information, which allows the industries or individual users to make critical live dependent decisions. Most of these IoT devices have resource constraints such as low CPU, limited memory, and low energy storage. Hence, these devices are vulnerable to cyber-attacks due to the lack of capacity to run existing general-purpose security software. It creates an inherent risk in IoT networks. The multi-access edge computing (MEC) platform has emerged to mitigate these constraints by relocating complex computing tasks from the IoT devices to the edge. Most of the existing related works are focusing on finding the optimized security solutions to protect the IoT devices. We believe distributed solutions leveraging MEC should draw more attention. This paper presents a comprehensive review of state-of-the-art network intrusion detection systems (NIDS) and security practices for IoT networks. We have analyzed the approaches based on MEC platforms and utilizing machine learning (ML) techniques. The paper also performs a comparative analysis on the public available datasets, evaluation metrics, and deployment strategies employed in the NIDS design. Finally, we propose an NIDS framework for IoT networks leveraging MEC.

## 1. Introduction

The Internet of Things has experienced tremendous growth in area-specific applications such as healthcare, smart transportation systems, smart agriculture, and industries to improve socio-economic development in recent years [1]. These IoT systems are composed of many interconnected sensors, actuators, and varieties of network-enabled devices [2] that exchange different types of data through both the Internet infrastructure and the private networks. The cisco research team predicted an average of 75.3 billion actively connected IoT devices by 2025 [3,4]. The absence of human intervention in data exchange between IoT systems makes it unique from traditional Internet technology. Growth in IoT devices has also increased the data network bandwidth demands. However, most IoT devices have resource constraints, making it challenging to execute the traditional security methods for system protection against cyberattacks. Critical concerns about the IoT device arise when there is a need to process sensitive information. Hence, it is essential to introduce the MEC platform that permits computation to be performed at the network end to address the resource-constraint problems in IoT systems [5,6]. MEC allows IoTs to offload high computational-intensive tasks to the proximal edge server [7].

Since the IoT has become the driving force of the current industrial revolution and the system for collecting live dependent data, it is vital to take cyber-security seriously [8,9]. Hence, there is the need for a Network Intrusion Detection System (NIDS) that can detect current and future attacks to protect the IoT network and the systems built on it.

### 1.1. IoT Network Intrusion Detection Systems

NIDS monitors the internet traffic across the devices in an IoT network. It acts as a defense line [10], which can identify risks and protect the network from intruders and malicious attacks.

NIDS is the primary tool used to combat network intrusion and various attacks in current computer network systems. NIDS examines and investigates the hosted device’s network, user actions, and discovers signatures of known threats and unknown malicious attacks within a network. Another goal of the NIDS is to monitor the IoT network, discover unauthorized intrusions within the network, and enable context-awareness to other devices connected to the network and the execution of necessary defense (firewall rules). NIDS also generates an alert or creates attack flags when it discovers internal and external attacks. Cyber-attackers initiate internal attacks through the compromised IoT devices connected to the network. Third parties outside the leading network initiate external attacks. There are principally three general elements of NIDS [11], namely Observation, Analysis and Detection. The Observation module monitors the network traffics, patterns, and resources. Analysis and Detection are normally the core part of NIDS. They can detect intrusions based on given algorithmic rules. The alert module raised attack flags if an intrusion is detected [11]. Considering that the development of NIDSs for IoT systems represents a significant challenge for information security researchers, this survey has covered the following topics: detection method, NIDS placement strategy, security threats, and validation strategy.

### 1.2. Motivation and Contribution

Several surveys have been published in diverse research areas related to IoT security, including security frameworks [12], security in context of eHealth [12], privacy issues [13], state of the art and security challenges [14], models, techniques, and tools [15], and attacks [16]. Some of these papers were published during the early stage of IoT system evolution. This paper focuses on the use of ML techniques to enhance security in IoT. Additionally, we investigate how the emergence of MEC can aid to develop a sophisticated security system (NIDS) for IoT systems. Moreover, this paper adds to knowledge of the comprehensive reviews of state-of-the-art design of NIDS for the resource-constraint IoT using the MEC, the implementation strategies, and the IoT dataset used. The study extends the design approaches used by researchers and how the proposed methods fit into NIDS design for IoT systems and MEC environment. We also proposed an NIDS framework for the IoT utilizing MEC architecture and demonstrated the possible ways to choose NIDS for the IoT devices based on some conditions. After reviewing the related research publications in this field, this work serves the audience from the following perspectives:We examine some of the leading IoT challenges presented in the recent research trends. Moreover, we analyze the feasibility of emerging technologies such as MEC to design security applications for the IoT.We investigate the ML-based NIDS designed for IoT and their implementation strategies.We examine the placement strategies used in MEC to develop NIDS for the IoT systems.We review various datasets and metrics used to develop NIDS for IoT systems.We propose a security framework for the resource-constrained IoT systems by utilizing the modern MEC architecture.

### 1.3. Paper Outlines

We have organized the rest of this paper as follows: Section 2 provides a summarized explanation of the key terminologies and security technologies used in IoT NIDS design. We classify the IoT security threat classes in Section 2 and introduce the fundamental security routines with attention to NIDS diversities. Section 3 describes the background of NIDS in IoT systems. This section also elaborates on the various types or categories of NIDS. In Section 4, we discuss how MEC are used to create NIDS, its design process, deployment strategies, and implications toward the heterogeneous IoT systems. Section 5 examines the mode of implementation of NIDS in IoT, while Section 6 examines the NIDS design strategies. Then, we compare and evaluate the various implementations of ML-based NIDS, datasets, and metrics used in NIDS design for IoT systems in Section 7. Finally, the survey concludes with an outline and future directions in Section 8. Figure 1 shows the brief taxonomy of the survey. Table 1 also provides some of the commonly used acronyms in this survey.

## 2. IoT System and Security Issues

The IoT system has evolved with many use-cases. Each of the applications of the IoT system comes with its own adverse security issues. This section elaborates on the IoT system and the related security problems.

### 2.1. IoT System

IoT systems consist of interconnected computing devices [17,18], mechanical or any object equipped with unique identifiers and the ability to transfer data over a network without requiring human-to-human or human-to-computer interaction [19]. IoT systems use distinctive network address schemes to connect to various objects or things [20]. Most IoT devices connected to the Internet are exposed to cyber-attacks due to the weak security systems as a result of resource constraints. Most IoT systems however operate autonomously via untrusted network connections and the Internet, which is also a contributing factor that exposes the network to cyber-attacks [21]. Considering cyber threats and network attacks versus the promising future of IoT systems, the security issues should be addressed with urgency.

### 2.2. IoT-MEC Architecture

There is no one defined architecture that is generally accepted by the IoT community. Different architectures are designed based on the use case, the technology used, and the size of the IoT network. In smart health use cases, refs. [22,23,24,25] proposed a smart architecture to monitor and track patients’ health records and transmit physiological parameters to control centers where the advance interpretation of the data is performed. Wang et al. [26] proposed a novel IoT access architecture based on field-programmable gate array (FPGA) and system on chip (SoC), which provides a unified approach to the IoT for a wide assortment of low-speed and high-speed devices with associated extendibility and configurability [27]. In technology and communication, Shang et al. [28] analyzed the security and challenges in implementing TCP/IP technology for IoT systems architectures. In [29,30,31], the authors also created different routing protocol for IoT communication. These architectures can be implemented in Autonomous Systems of Things (ASoT) and legacy Autonomous Systems (ASs). Different configuration techniques were applied for IoT routing architectures. Notwithstanding, different architectures have been implemented for assorted use scenarios such as in [32], where the authors proposed an IoT-based architecture for sport (football). Their aims were to embed sensing devices (e.g., sensors and RFID), telecommunication technologies (e.g., ZigBee) and cloud computing with the mission to monitor the health of footballers and eradicate the occurrence of diverse health problems in football. In all the above-mentioned architectures, they have different use-cases; therefore, the sensitivity of data in each scenario differs. Creating security for each system requires special knowledge in the field of application. However, the IoT-MEC architecture follows the level such as Sensor Level (IoT end-device), MEC/Fog Layer, and the Cloud. Detailed the structure of the IoT devices, MEC, and the cloud is shown in Section 3.

#### 2.2.1. Sensor Level (IoT End-Device)

Mobile devices (e.g., smartphones, tablets, and laptops) and IoT devices (e.g., industrial actuators, wearable devices, and smart sensors) make up the bottom of the IoT or sensor layer [33]. Based on their ownership, co-location, and use cases, these devices can be grouped into clusters.

#### 2.2.2. MEC/Fog

The MEC is a layer that lies between mobile devices and the cloud [19]. The MEC system directly connects to the IoT gateway. The MEC is usually equipped with high-performance computational resources [34] to support the IoT devices connected at the network edge. In an IoT MEC-enabled environment, security is vital. Hence, the computational resources of the MEC provide an opportunity to design a sophisticated security system to protect the IoT system.

#### 2.2.3. Cloud

The cloud layer is part of the core network, and it consists of various cloud servers and data centers that can process and store massive quantities of data [33].

### 2.3. IoT Network Security and Attacks

IoT applications and technologies are expected to step farther ahead than anyone could possibly imagine [35]. However, the development of IoT technologies is still in transition and has not fully matured in terms of security protection. The IoT systems have several security challenges [16], such as non-uniform manufacturing standards and update management problems caused by IoT software developers. Physical handling of the security issues and users’ ignorance due to their lack of awareness of security problems associated with the IoT systems are also critical problems. Moreover, there is no standard defined security architecture accepted by the IoT community. Different security architectures are adopted based on the use case, system requirement, the available technology, and the IoT network size. For example, in the intelligent health use cases discussed in the research works in [22], the authors proposed different intelligent security architectures to monitor and track patients’ health records. The sensitivity of data in IoT use cases varies. Hence, the need to create security for each scenario requires special knowledge in the field of application. It is therefore convincing that NIDS designed to target the various architecture and use case in IoT requires customization [36]. Apart from adopting encryption methods to safeguard the IoT system data transmission, the network and environment of the IoT systems must also be protected. Unfortunately, traditional network security systems are not applicable in IoT systems due to the nature of the resource constraints. Furthermore, different network attacks have also emerged due to the rapid development and wide application of the IoT system applications [16,37]. The amount of attacks will continue to rise as the IoT use cases expands. Being able to recognize and comprehend the intense rise in cyber-threats in the IoT system drastically decreases the risk of a network security and data breach. Detailed discussions of these attacks are found in [4,16]. Recently, some of the most frequent attacks initiated to target the IoT systems include:
Spoofing: Attackers impersonate a legitimate IoT system in a network to gain control or illegal access to the network. When the attacker obtains access, they initiate DoS and Man-In-the-Middle attacks against targeted devices [38,39,40].Denial of Service (DoS): A cyber-attack makes IoT systems or resources on the network unavailable to the intended legitimate users. These attacks are launched by the perpetrators to temporarily or permanently disrupt a host IoT system’s services [41,42,43].A distributed denial-of-service (DDoS): The attack is a malicious network attack that can disrupt the regular traffic and the network’s services. DDoS floods the target or surrounding infrastructure with highly intense network traffic [44]. DDoS attacks become successful when attackers apply multiple compromised computing systems as the sources to generate a large amount of network traffic. The targets can be either the IoT system or other networked devices [45,46].Jamming: Most IoT devices communicate to other devices through wireless networks. The perpetrators attack the targeted IoT system and send a fake signal to interrupt the radio transmission, thereby depleting the bandwidth, processing power, and memories [47,48].Man-in-the-Middle: Figure 2 shows a diagram of Man-In-the-Middle (MITM) attacks. MITM attackers secretly relay and manipulate the communication between two IoT systems, the remote devices, and eavesdrop the private communications among the parties [49,50].Privacy leakage: Data protection and privacy should be a priority in IoT system communication. Most IoT system manufacturers and users pay minimal attention to the contents of information stored on their devices and how third-party companies analyze and process such IoT privacy information. IoT systems such as wearable devices that collect user’s information about their location, situation, and health information have contributed immensely to the high rise in the risk of personal privacy leakage [51,52].Marai Botnet Attack: Mirai is malware that turns networked devices into remotely controlled bots that cyber-attackers use as part of a botnet in a large-scale network. It primarily targets online consumer devices such as IP cameras and home routers. Attacks such as DoS/DDoS also used Mirai as a prevalent initiator [53,54].Sybil Attack: Sybil attacks are found in peer-to-peer networks. A Sybil attack subverts the IoT device’s identity to create many anonymous identities and consume a disproportionate power. It is named after Sybil, the author of the book Sybil, which is a case study of a woman dealing with a dissociative identity disorder. An IoT device in a network that operates multiple identities often undermines the authorized network access in reputation systems [55,56]. Sybil attacks capitalize on this weakness in the IoT system network to initial attacks.AI-Based Attacks: AI attacks have been around since 2007 [57,58], and the high threats they pose to the IoT are becoming more prevalent. Hackers now create AI-powered tools that are quicker, simpler to scale, and more efficient than human-centric attacks. This is a significant danger to the IoT ecosystem. While the techniques and features of conventional IoT risks offered by cybercriminals may appear to be the same, the scale, automation, and customisation of AI-powered attacks will make them increasingly difficult to eradicate.

Table 2 summarizes some common attacks in IoT systems and their modes of initiation.

## 3. Background of NIDS in IoT and Their Classification

The IoT security problems are similar to the early genealogy of computers in the mid-1990s [59] when the unreliability of personal computers (PC) security required emergency attention. Many computer networks were vulnerable to threats, and there were no direct approaches to fix them. Organizations kept vulnerabilities as a mystery and did not discharge security overhauls rapidly. Moreover, it was hard, if not incomprehensible, to motivate clients to introduce new security systems. In recent years, this has changed because of a mix of full divulgence distributed vulnerabilities to organizations’ constraints. The skills of attackers, the developing technologies, and the enormous growth of Internet traffic have made it difficult for any existing NIDS to offer a reliable service to the IoT system. However, a close examination of the IoT system’s cyberattack shows that there usually exists a behavioral pattern in the attacks [60], which can be learned when ML and MEC are combined to design NIDS more effectively.

This section provides a comprehensive review of the various classifications of the state-of-the-art NIDS used in IoT systems. Since IoT networks operate similar to traditional general purpose computer networks, researchers have adopted the traditional techniques to design NIDS and propose numerous models. We based our classification in this section on the framework, implementation, and operation to categorized NIDS into Signature, Anomaly, Specification, and Hybrid based [61].

### 3.1. Signature-Based NIDS in IoT Systems

IoT devices have limited resources, such as the memory, the processing ability, and the energy constraints of the physical IoT devices [62]. Therefore, the use of traditional signature-based intrusion detection systems on these devices is practically impossible. To construct robust detection systems for IoT, signature-based NIDS often requires large datasets. However, conventional signature-based NIDS must be restructured in accordance with the resources of the IoT devices. Various attempts have been made to design signature-based NIDS for IoT devices. Kasinathan et al. [63] proposed a signature-based NIDS framework. Their framework identifies DoS attacks in 6LoWPAN-based [64] networks. The authors used signature-based NIDS known as Suricata (an open-source NIDS software) [65] to test their proposed framework. Suricata was designed to detect intrusion in general computer networks, which does not specifically target IoT networks. Moreover, there were no clear evidence of the effects of Suricata on their IoT devices use in their research. In [66,67,68,69], the authors presented different strategies to design NIDS to protect the IoT devices. The authors used ML methods to design signature-based NIDS to detect network attacks in IoT devices. In [70,71], the authors applied deep reinforcement learning to create NIDS for industrial IoT. This approach combines the observational capabilities of deep learning with the decision-making capabilities of reinforcement learning to allow the effective detection of various types of cyber-attacks on the Industrial Internet of Things. However, in each of the research works, the experimental results presented are promising, but the authors failed to demonstrate how their method could work in a real-world IoT network environment.

### 3.2. Anomaly-Based NIDS for IoT Systems

Anomaly-based NIDS compare an IoT system’s activities instantly against the standard behavioral profile and generate notifications whenever a deviation from such operation exceeds a threshold. This approach is efficient in detecting new attacks on IoT systems. In particular, those attacks initiated misuse of the IoT devices’ resources. Unlike signature-based NIDS, anomaly-based NIDS are designed with either normal or anomaly data (one-class of data) [72]. Several NIDS developed to secure IoT devices use anomaly-based techniques since they can be modeled with the intention of being lightweight. In [73], the authors highlighted the necessity to obtain custom anomaly-based security in an IoT system. They also emphasized that vulnerabilities in IoT systems are due to insecure web interfaces, insufficient authentication and authorization, insufficient transport layer protection, broken cryptography, insecure software/firmware updates, or poor physical security. Their research presented an efficient hierarchical anomaly-based intrusion detection system. In their experiments, the proposed framework applied a resilient policy that enables the IoT system to detect malicious attacks. In [74,75,76,77], the authors also used ANN to detect intrusion in the IoT system’s gateway. They used ANN to detect anomalies in the data sent from the edge devices. The researchers connected multiple IoT devices and a high-resourced network device to operate as the gateway. Their results were immaculate and promising. In [78,79,80], the authors presented an NIDS algorithm that detects attacks in IoT systems, which uses anomaly detection systems based on different ML methods. The authors explained that an IoT network attack typically leaves its traces on the system. Using this approach, the authors proposed three techniques to identify such anomalies in their network. However, they did not demonstrate any experimental results of false-positive rates, which is a major problem with anomaly-based NIDS. The authors also performed a further study on the power and memory consumption of IoT devices. The authors of [81,82] proposed a smart home NIDS that alters the decision function of its underlying anomaly categorization models autonomously in response to changing conditions in the smart home.

The research works in this subsection demonstrate that anomaly-based NIDS often start by establishing a baseline of the network’s normal traffic and activity. They compare the current condition of network traffic to this baseline in order to find patterns that deviate from the normal traffic. Anomaly-based NIDS are suitable for IoT systems due to the ability to scale them into a lightweight but are prone to false-negative rates. The false-alarm rate is due to two main reasons: (1) researchers failed to obtain a credible training dataset that holds all the necessary intrusion signatures; (2) not all normal or intrusions behaviors were captured during training.

### 3.3. Specification-Based NIDS for IoT Systems

The Specification-Based NIDS defines a set of rules that can be manually updated by the IoT systems administrator to detect intrusion on a network. The specification-based approach identifies threats by the definition of high-level rules in the IoT network environment, operational features, and mechanisms. Due to the resource constraint problems in IoT systems, specification-based NIDS is recommended. However, specification-based NIDS detects only specific kinds of attacks that fall within the rules defined in IoT systems. The critical difference between specification-based and anomaly-based NIDS is their manually defined rules for each attack [83]. The authors argued in [84,85] that specification-based NIDS can identify attacks that disturb the optimal protocol structure of IoT networks. The authors presented an RPL specification to be used as a reference to evaluate the IoT node’s behaviors, which was created using a semi-auto profiling approach that creates a high-level abstract of activities using network simulation traces. This specification, which includes all valid protocol states and transitions as well as statistics, is implemented as a collection of rules in the NIDS agents, which are transmitted in the form of cluster heads to monitor the whole network. Instead of allowing the cluster head (usually an MEC server) to monitor everything, the members of the cluster report relevant information about themselves and their neighbors to the cluster head to conserve resources. Moreover, the authors in [86,87,88] applied a specification-based method to detect attacks in the IoT network. All the research works in this subsection demonstrates that specific attacks such as those that target RPL protocols in IoT devices can efficiently be detected by applying specification-based NIDS.

### 3.4. Hybrid-Based NIDS for IoT Systems

The hybrid strategies use two or more signature-based, anomaly-based, and specification-based NIDS to create one system. Hybrid-based NIDS for IoT systems capitalize on the strength of the combined methods and minimize the impact of their setbacks. To eradicate false alarm rate flops in anomaly-based NIDS and keep the advantages of signature and specification-based detection, some researchers proposed hybrid-based NIDS for IoT systems. Based on this combined approach, we categorize the hybrid NIDS into two categories:Sequence-based: either anomaly or misuse detection is applied first, and a different technique is applied next.Parallel-based: multiple detectors are concurrently applied, and the final decision is made based on multiple output sources.

The most common type of hybrid NIDS combines signature-based detection and anomaly detection. In such a hybrid system, the signature-base detection technique detects known attacks, and the anomaly detection technique detects novel or unknown attacks. Due to the resource constraint nature of the IoT systems, applying hybrid-based NIDS in the same IoT systems is practically difficult and not recommended. However, a lot of research works demonstrate the feasibility of hybrid-based NIDS on IoT systems.

According to Ning and Lui [89], human activities also cause security problems through the architecture differences and the standards used in IoT design. The authors introduced a hybrid security framework called IPM (that deals with information, physical, and management security perspectives). Cervantes et al. [90] combined Watchdog, Reputation, and the Trust Strategy to detect intrusion in IoT systems. Their approach detected the sinkhole attack. INTI (Intrusion detection of Sinkhole attacks on 6LoWPAN for Internet of Things) investigates NIDS’s effects that cause problems with the performance. According to their study, the sinkhole has advanced effects on networks such as mobile ad hoc networks (MANETs), wireless sensor networks (WSNs), and vehicular ad hoc networks (VANETs). They emphasized that sinkholes cause a high positive rate, false negatives, high energy consumption rates and slow down the IoT system’s performance. Their system analyzed each IoT device’s behavior in the network and checked for the existence of sinkholes. They also identified attacks on IoT devices through inter and intra-cluster communication channels. Constrained Application Protocol (CoAP), designed by Krimmling and Peter [91], used hybrid NIDS to combat intrusion in IoT. Their system was tested with the OMNET++ simulator. The authors also evaluated their framework with an intelligent transport application system that uses CoAP. The heterogeneous nature of the communication network poses a critical challenge to IoT systems. Therefore, they employed signature-based and anomaly-based NIDS in their research. They detected known attacks with signature-based NIDS, while the anomaly detected unknown attacks that they introduced during their experiment.

The application of hybrid-based NIDS in IoT systems solves the problems associated with a single NIDS. However, hybrid-based NIDS require more computational resources, storage capacity, and energy, which makes them an imperfect fit for the IoT system. Recently, researchers have utilized MEC computing to resolve the resource constraint problems associated with the IoT system. Researchers apply MEC technology to design NIDS for the IoT to offload the most resource-intensive part of the NIDS to a proximal MEC server [92,93]. A detailed discussion of this approach is provided in Section 4 of this paper.

### 3.5. Host and Network-Based NIDS for IoT Systems

NIDS design for IoT can also be classified based on their mode of operation. There are two main mode of NIDS operation: namely, Host-based and Network-based.

Host-based NIDS identifies the threats and attacks on the same system or devices in which they have been installed. Host-based NIDSs use system data from normal routine processes, the same subnet as the monitored machines, and system call on a particular device [94]. Wang et al. [95] designed a host-based NIDS that detects cyber attacks in IoT. The authors experimentally demonstrated the existence of the mimicry attacks and demonstrated the need to prevent them. In [96], the authors propose a host-based intrusion detection system that was created and prototyped to safeguard IoT devices, that make up IoT network backbones. In their approach, a set of suggested NIDS performs traditional security verification, and it engages with a controller from the host-based NIDS to enable the coordination of intrusion detection activities in response to IoT devices affected by DDoS attacks throughout the network. The of authors [97] argued that network-based NIDS required a large amount of data to make meaningful decisions. Table 3 shows some differences between Host-based and Network-based NIDS.

## 4. Multi-Access Edge Computing as a Resource to Provide Security for IoT

The interest in applying MEC over cloud computing to combat network attacks in IoT systems has risen dramatically recently. Hence, MEC standardization has become of pivotal interest to major telecommunication and network stakeholders under the supervision of institutions such as the European Telecommunications Standards Institute (ETSI) and Open Edge Computing Initiative (OEC [98]) [7,99,100]. Moreover, it is known that NIDS designed for IoT systems use case scenarios based on MEC require high quality of service, reduced latency, high throughput, and real-time operation. Researchers desist from the use of cloud computing due to its crucial drawback of low propagation delay (high latency) [101,102]. The most influential features that make IoT system developers select MEC over cloud computing for NIDS design include:Providing security context-awareness (the capability of the MEC server to disseminate real-time security information to the IoT device) [103];Energy-saving during and after data transfer;Improvement of privacy/security in IoT (users are deprived of the total ownership of their data in cloud computing, which results in private data leakage and loss) [104];Optimal resource allocation by the MEC for the IoT system [105].

Figure 3 shows the structural layout of a cloud or remote server, the MEC, and the IoT.

MEC-based NIDS for the IoT system is implemented based on virtualized platforms that use modern technology improvement in network functions virtualization (NFV) [106], information-centric networks (ICN) [107], and software-defined networks (SDN) [108,109,110]. Specifically, NFV enhances an edge device to supply computing services to numerous connected IoT systems to create multiple virtual machines (VMs) to execute their NIDS tasks spontaneously or operating different network functions [111]. The IoT systems offload computationally intensive tasks to the MEC platform for execution. The security task offloading takes place in different forms. The next sections examine the various possible offloading techniques for NIDS implementation in IoT systems.

### 4.1. NIDS Task Offloading in IoT-MEC Environment

This section elaborates the mechanism of slicing the NIDS (tasks) and transferring part or all to the MEC platform. Task offloading is the primary technology used in MEC and Fog for such kind of NIDS design to secure the IoT system from cyber attacks. Task offloading is employed as a mechanism to reduce energy consumption, latency, and the resource-constraints problems in IoT system [112,113,114]. Hence, the proximity of the IoT system from the MEC server during NIDS task offloading is significant. It determines the intensity of tradeoffs (bandwidth, amount of energy, scheduling algorithms, etc.) required for a complete execution cycle of the NIDS in IoT systems [115,116]. NIDS tasks offloading can be categorized as follows: Binary NIDS Tasks Offloading; Partial Computational NIDS Tasks Offloading.

#### 4.1.1. Partial Computational NIDS Task Offloading

Partial Computational Offloading dividends the underlying NIDS into sub-tasks. The IoT system offloads sections of the sub-tasks to a proximal edge computational platform for execution, while less computational tasks are executed on the IoT system. Wang et al. [117] proposed a distributed framework in their NIDS design for IoT systems. Their approach uses the partial computational offloading technique to distribute the NIDS on the IoT system and MEC platform. They also believed that the NIDS sub-task offloading to the MEC platform reduced workload and energy consumption. In [80,118,119], the authors proposed an internal distributed NIDS for IoT. They offloaded sections of the NIDS system across the various IoT systems. Each IoT device in the network monitors its internal network but provides computational support to other devices whenever there is an intrusion. Their analysis showed that the above approach has higher accuracy of detection compared to a standalone NIDS. Figure 3 shows the structure of an *MEC* design. Different partial computational offloading algorithms have been designed for IoT systems in a 5 G heterogeneous network. A review of these research works shows that the algorithms implemented are directed toward minimization of energy consumption and low latency [120,121,122]. However, the researchers pay less attention to the use of MEC offloading to secure the IoT. In [123], the authors implemented a multi-user partial computation offloading. To this aim, Alladi et al. [123] described a Deep Learning Engines (DLEs)-based artificial intelligence (AI)-based intrusion detection architecture to identify and classify vehicle traffic within internet of vehicles (IoV) networks into probable cyber-attacks. These DLEs were also implemented on MEC servers rather than remote cloud, which takes into account the mobility of the vehicles and the real-time requirements of the IoV networks. The following conditions must be considered to apply partial computational offloading to design NIDS for IoT systems:First, the sequence of execution functions or routines cannot be discretionarily selected because the outputs of some components of the NIDS model are the inputs of others.Second, due to resource constraints and security considerations, some algorithms or routines of the NIDS must be offloaded to the MEC for execution, while others have to be executed locally.

#### 4.1.2. Binary NIDS Task Offloading

The binary NIDS task offloading allows the IoT device to execute the entire NIDS either on the local IoT device or remotely in the edge platform [124]. An integrated NIDS cannot be divided into sub-tasks. Hence, those kinds of NIDS are executed fully in the edge platform or locally in the IoT system. In [125], the authors proposed a Markov decision process as a means to enhance tasks offloading to the edge. They first proposed a double deep Q-network (DQN)-based strategic computational offloading algorithm to study the optimal policies before they implemented their original technique.

### 4.2. MEC-Based NIDS Design for IoT Systems

In NIDS design, the MEC provides the hosting infrastructure to support the computational intensive security services that manage the real-time detection of threats within and outside the IoTs’ network. Table 4 summarizes the various research works that used MEC technology to design NIDSs for IoT systems. In this section, we review the use of MEC platforms to design NIDS for IoT systems and discuss their strengths and weaknesses. Since there are limited MEC-based NIDS for IoT systems, we provide a comprehensive survey of the state-of-the-art related designs and their application.

In [105,126], the authors provided closely related findings on MEC-based NIDS for IoT systems. Their research provided a security end-to-end protocol for resource-constrained devices, especially in the context of healthcare sensors. The authors also applied security functionality and reduced computationally intensive operations to satisfy the IoT system’s resources constraints problem. The authors offloaded the computationally intensive algorithm of the NIDS to an edge platform in neighboring trusted devices. They introduce a technique to select the proximal MEC platform with unique selection criteria. Ranaweera et al. [127] highlighted some security flaws in 5 G-based IoT use cases that have been implemented in the MEC setting. They suggested solutions to mitigate the security flaws that they identified. In [128,129], the authors proposed security as a service (SECaaS) architecture that protects the IoT from cyber-attacks. They test their suggested architecture by creating a virtualized infrastructure that combines lightweight and hypervisor-based virtualization technologies to design a security system for the IoT using MEC. Sha et al. [130] proposed a new security system deployed at the MEC platform to secure connected IoT systems. Their security system, which was known as EdgeSec, comprised six major modules that systematically handle specific security challenges in IoT systems. The modules include a security profile manager, security analysis, protocol mapping, a security simulation, communication interface management, and request handling. In their experiment, each IoT device is registered to the security profile to manage the device-specific information collected and the device-specific security requirements. An NIDS deployed to an MEC platform controls the security of the various IoT systems. They also utilized protocol mapping to choose the appropriate security protocols from the protocol library for each IoT device in their setup. In [131], the authors proposed a privacy-aware scheduling algorithm based on MEC. The research’s primary purpose is to execute tasks from different IoT systems with different privacy requirements connected to different MEC platforms. The proposed scheduling algorithm also aims to satisfy the real-time requirements of IoT system operation. The authors of [105] in their paper examine ways to improve the security of MEC systems by utilizing the cooperative mechanism amongst non-orthogonal multiple access (NOMA) user pairs. They presented a two-slot hybrid cooperative NOMA security (THCNS) system that uses cooperative interference between NOMA user pairs to improve the security of offloading, taking into account the varying latency needs of IoT users.

The research works in this section demonstrate that the use of MEC to create NIDS for the resource-constraint IoT devices is still in the early stage. Hence, researchers must provide robust frameworks for the offloading process, detection models, and implementation strategies to design sophisticated NIDS to secure the IoT system and their network environments.

## 5. Mode of NIDS Implementation in IoT Systems

Additionally, we categorize the NIDS based on their mode of implementation. This category of NIDS depends on the location, position, and part of the network where it is hosted. This subsection discusses the NIDS placement strategies in IoT systems.

### 5.1. IoT NIDS Based on Placement Strategies without MEC

There are different techniques to implement NIDS in IoT systems. Most of the implementation strategies depend on the structure and the type of network technology available. Although different IoT systems have different network architecture, however, we classified the placement strategy under the following:Physical domain: interfaces the sensors to the IoT board.Network domain: gives the IoT device the capabilities to transfer data.Application domain: gives the user power to interact with the IoT device.

Usually, NIDSs are placed in network equipment such as routers, installed in between the IoT device network and the Internet connectivity to the Internet Service Provider (ISP). These routers are known as border routers [132]. The following subsections explain some of the NIDSs’ placement strategies in IoT systems.

#### 5.1.1. Centralized Placement

In the centralized placement, NIDS analyzes all the traffic that goes in and out of the boarder router from the connected IoT devices. The centralized placement strategy has the disadvantage of not identifying attacks within the internal IoT networks. In [63], the authors used the centralized placement method to deploy their proposed NIDS. Their research focused on prevention of the DoS attack against the IoT system. Therefore, they used a dedicated host that sniffs network data and analyzed it. They connected the host NIDS through a wired network connection, leaving the remaining IoT devices connections in a wireless mode. The technique helps the NIDS to identify the DoS attack when the network is compromised. Wallgren et al. [133] designed a centralized NIDS placement system by setting up their NIDS in a border router. Instead of analyzing the network traffic, echo messages were sent to the various IoTs regularly to determine whether the network has been under attack.

#### 5.1.2. Distributed Placement

In the distributed placement strategy, the NIDSs are spread across the various IoT devices in the network. Due to the IoT system’s resource-constrained nature, each system must be optimized independently, and the NIDS must be lightweight.

IoT devices that are configured to keep track of threats in nearby connected devices are known as watchdogs. Cernantes et al. [90] used this placement strategy in their work. The INTI used the watchdog to detect and mitigate attacks. In their experiment, an IoT device within the network clusters was classified as the leader or the head node. Each connected IoT device’s role can change over time due to the network reconfiguration policy they introduced. Whenever an IoT device detects an attack, it broadcasts a message to the other connected devices to protect themselves from the attacker.

#### 5.1.3. Hybrid Placement

The hybrid NIDS placement integrates the technique of centralized and distributed placement strategy. It dwells on the advantages of both methods and avoids their pitfalls. In Pongle and Chavan [134], the authors stated that IoT devices are responsible for detecting changes in their neighborhood and sending information about neighbours to centralized modules.

Their experimental results showed that the energy, computational, and memory usage overhead matched the constrained IoT system’s resources. In [118], the authors proposed an NIDS that allocates different responsibilities to the border router and the connected IoT systems. The NIDS module, in their experiment, monitored the neighbors to identify possible intrusions. When an intrusion is detected, the IoT sends an alert to the NIDS model placed within the border router. In [135,136], the authors used a distributed placement strategy to implement their proposed security system for the resource-constraint IoT system. Thanigaivelan et al. [118] classified their system as a distributed NIDS. However, the border router’s central role to make the final decision about intrusion detection makes the proposed NIDS a hybrid approach.

### 5.2. Distributed Placement Strategies of NIDS in IoT with MEC

*Distributed NIDS* (DIDS) consists of two or more NIDS places over a wide span of a network(s), which interact with each other. Some distributed systems have a centralized NIDS that supervises the other security systems placed in other IoT systems in the same network [137]. DIDS operates by the process of cooperative intelligent systems spread across the network in a distributed IoT MEC environment. Due to the resource constraints in IoT, distributed data-flow programming models are used to build IoT applications utilizing the MEC platform [138,139].

In [140], the authors proposed a novel NIDS architecture model for the IoT system. The model operates based on the MapReduce approach in the context of distributed detection. The model also incorporated a multi-faceted detection technique based on anomaly-based and misuse-based NIDS agents. Farhoud et al. [141] designed a distributed and lightweight NIDS based on an Artificial Immune System (AIS). They distributed their AIS across the cloud, the edge, and the IoT devices. In [142], the authors developed a distributed NIDS that detected attacks, where part of the detection model is placed on the IoT device and hosted the remaining model on the resources of the Internet Service Provider (ISP). They designed the NIDS to secure home gateway devices that connect smart home IoT devices to the Internet. Zeeshan et al. [143] designed and evaluated an NIDS system suitable for small IoT devices. They used a trust management process that allows IoT devices to handle vital information about connected neighbors. This mechanism allows the IoT device to identify malicious patterns in the network. [144] proposed an NIDS that utilizes information flow processing to obtain event data from distributed sources as soon as relevant data arrive. Their system also was equipped with capabilities to detect attacks in real time. In [145], a collaborative NIDS was proposed by distributing the NIDS algorithm across the various IoT devices to detect intrusions. In this way, the IoT devices share the cost of running the NIDS, which reduced the energy, processing power, and storage capacity required for the detection.

### 5.3. Centralized NIDS Placement Strategies Based on IoT with MEC

In the centralized strategy based on the MEC platform, the NIDS is placed in the central part (proximal MEC platform). Usually, most network system administrators adopt centralized placement strategies and install NIDS in the border router. However, analyzing the network traffic that crosses the border router is insufficient to detect attacks that involve IoT devices within the network. Then, researchers must create NIDS that can monitor the traffic transferred between IoT devices. Cho et al. [146] proposed NIDS for IoT systems to analyze the packets that traverse through the border router (proximal MEC platform) to connected physical IoT devices and other connected network devices. The work focused on botnet attacks that pass through the border router. Wallgren et al. [133] proposed a centralized NIDS IDS that was placed in the border router. Their main aim was to detect network attacks within the IoT network. The authors proposed a heartbeat protocol placed in the border router that transmits ICMPv6 echo requests to all the IoT devices in the network to detect attacks. Jun et al. [147] also proposed Complex Event-Processing (CEP) techniques for network intrusion detection in IoT systems. The authors adopted a centralized approach and placed the NIDS in the border router to monitor network traffics. The main advantage of the proposed system is the use of features of the events flows to detect the intrusions, which can reduce the false alarm rate. Table 4 summarizes some of the NIDS designed with MEC by the research community.

**Table 4 sensors-22-03744-t004:** Analysis of NIDS in IoT systems using MEC and Placement Strategies.

Reference	Detection Methods	Threats Identified	Placement Strategies
Thanigaivelan et al. [118]	Network fingerprinting	Network Anomalies	Distributed locally in the IoT Devices
Ferdowsi and W. Saad [119]	Artificial Neural Network (ANN)	Data Intrusion	Distributed locally in the IoT Devices
R. Chen, C. M. Liu, and C. Chen [148]	Artificial Immune-Base (ANN)	Network Intrusion	Distributed locally in the IoT Devices
Wang et al. [117]	ZeroR, KNN, SVM, NaiveBayes, Neural Network	Network Intrusion	Distributed with MEC
Hosseinpour et al. [141]	Negative selection algorithm	Network Intrusion (DoS/DDoS)	Distributed with cloud, Fog, and MEC
D. A. Abeshu and C. Naveen [149]	Stacked Autoencoder Deep Learning	Network Intrusion Attacks	Distributed with MEC
A. Abeshu and C. Naveen [150]	Multi-Layer Deep Network	Network Intrusion Attacks	Distributed Locally and Fog
Diro Abebe and Chilamkurti Naveen [151]	LSTM Deep Learning	Network Intrusion Attacks	Distributed Locally and Fog
A. S. Sohal et al. [152]	Markov Model and Virtual Honeypot Device	Network Intrusion Attacks	Distributed with Fog to MEC
J. Stolfo et al. [153]	Offensive Decoy technology	Cloud Data attacks	Distributed with cloud and Fog
B. K. Sudqi et al. [154]	Multilayer Perceptron (MLP)	Network Intrusion Attacks	Centralized on Fog
Sheikhan Mansour and Bostani Hamid [140]	MapReduce approach	anomaly-based and misuse-based attacks	Distributed locally on IoT
Gajewski M et al. [142]	Co-responsible distributed NIDS	Network Intrusion Attacks	Distributed between IoT and ISP’s Server
Xingshuo et al. [155]	Sample extreme learning Machine	Network Intrusion Attacks	Distributed between IoT, MEC, and Fog

## 6. NIDS Design Strategies for IoT Systems

NIDS design strategies can be categorized into Packet Parsing-based, Payload Analysis based, and encrypted traffic analysis [156]. NIDS examines the payload information of transmitted packets in the IoT network. However, this strategy sometimes fails because the payload information is inaccessible in encrypted transmission. To solve the limitations, researchers apply encryption traffic analysis. Due to the resource constraint problems in IoT, most NIDS designed to protect IoT systems adopt lightweight design principles. This section examines the various design strategies and the technique of reducing the NIDS into a lightweight to match the resources in the IoT system.

### 6.1. Packet Parsing-Based NIDS Design for IoT

The network packets form the fundamental structure of network communication and are the data that are used for NIDS. They contain different forms of binary data that need first to be parsed [157]. Data transmitted through the network contain the header and the application data. The header consists of IP addresses, ports, and other essential fields specific to various protocols. The main advantages of using packets parsing to design NIDS data sources include:Packets consist of communication contents and can effectively be employed to detect U2L and R2L attacks.Packets contain TCP/IP data content and timestamps that precisely provide the NIDS information about attack sources.Packets analysis promote real-time data processing without content caching within the IoTs’ network.

Meanwhile, each of the packets do not indicate the complete communication state or every packet’s contextual information. Hence, it is difficult to detect some attacks, such as DDoS [158]. The packet analysis detection methods support packets that mainly include packet parsing methods and payload analysis methods.

Packet parsing-based detection utilizes different sorts of protocols in network communications such as HTTP and DNS. These protocols operate in different forms; the packet parsing-based detection methods primarily concentrate on the protocol header section. The standard practice is to extract the header fields using parsing tools (i.e., Wireshark, Scapy, or the Bro) and convert the most vital fields’ values into feature vectors. The header section also extracts necessary features from packets using classification algorithms to design attack detection systems. In [159,160], the authors proposed models to captured packets from a trustworthy enterprise network and parsed them with different tools. In packet parsing-based NIDS design, the packets are grouped into protocol types. Different clusters are then formed based on the information with the ML cluster algorithm using the protocol datasets. Thus, the first dataset contains many clusters, where the information from any given cluster is homologous. Finally, a new classified dataset is created for the NIDS model with an ML algorithm to detect IoT system attacks.

### 6.2. Payload Analysis-Based NIDS Design for IoT

Payload analysis-based detection is also another approach to design data-centric NIDS. The payload of the application layer protocols forms the data section of the packet sent through the IoT network. The payload analysis-based methods work well in a multi-protocol network environment because the packet headers do not need to be analyzed [161,162]. As a type of unstructured data, payloads can be processed directly by deep learning models [163]. However, this method does not include encrypted payloads.

### 6.3. Encrypted Traffic Analysis-Based NIDS Design for IoT

The accessibility of dataset features in encryption data is hardly limited. As a result, it is tempting to concentrate on assaults that are ostensibly apparent due to increased traffic. Scanning, brute force, and DoS/DDoS attacks are all examples of these types of attacks. Many research studies have demonstrated that such attacks may be seen in network traffic [164,165]. Other sorts of attacks on encrypted traffic that do not involve decryption, on the other hand, have received far less attention. The attack must meet a few characteristics in order to be apparent without decryption. To begin with, attacks can only be detected if traffic passes through the NIDS. Some Nattacks can be carried out locally or via a different path to the target. In certain instances, NIDS may not be able to detect the direct attack, but it may be able to detect the indirect attack. Second, the attack must alter some aspects of network flow. Without depending on existing attack signatures, zero-day and targeted cyber-attacks may be detected by concentrating on network traffic flow and modeling different detectable attack aspects [156].

### 6.4. Lightweight NIDS Design in IoT Systems

Due to the aforementioned resource constraints in IoT systems, the idea of lightweight NIDS is coming into the limelight in the research domain to conform to the IoT devices architecture. Based on Tiburski et al. [166] research, lightweight NIDS does not mean simplicity, but it should be able to deliver its mandated security duties without compromising the resource constraints. Sedjelmaci et al. [167] reflected on a lightweight NIDS in their research as a security system that is more resilient and robust but modest in size to enhance the ease of implementation on the IoT devices in a network. In [168], the authors also defined a lightweight NIDS as having the ability to save energy and to require less computational resources. Therefore, we define NIDS as lightweight when it can fully perform the required security operations regardless of the IoT device’s resource constraints. One way of achieving lightweight NIDS is to use feature selection and reduction models. Feature reduction such as PCA, auto-encoder, Pearson correlation, etc. [169] are used to reduce the dimensionality of datasets, which affect the model size, computation, storage capacity, and complexity of the NIDS model on the IoT device. Xingshuo et al. [155] proposed a lightweight ML algorithm called sample selected extreme learning machine (SS-ELM). They assumed that the IoT device could not host a large size of data from their analysis. Therefore, the data selection occurs on the server remotely before being submitted to the MEC for training and detection. To create lightweight NIDS for IoT systems, researchers use one or a combination of the following principles:Applying new protocols to decrease the number of computational operations required by a traditional NIDS;Utilizing optimization techniques to design a security system such that the scalar multiplication, addition, and doubling are reduced.Offloading of all or parts of the computationally intensive algorithms to a proximal device (MEC platform) in the same network that has more resources than the IoT device;Using modern algorithms that require less computational resources instead of the use of classical, alternative methods;Utilizing data feature dimensionality reduction. Different algorithms such as sparse auto-encoder, the Ranker method, principal component analysis (PCA), etc., are used by researchers to reduce the dimension of the dataset to lightweight ML-based NIDS for IoT devices.

All the above methods have effects on the performance of the NIDS. Therefore, extra care is required to minimize the loopholes for cyber-attackers to mount attacks against the IoT device.

## 7. Machine Learning, Dataset, and Metrics for IoT Systems NIDS

Regardless of the number of NIDS development years, current NIDSs face challenges in enhancing detection accuracy, reducing the false alarm rate, and detecting newly created attacks. To eradicate the above problems, modern researchers focused on developing NIDS that utilize ML methods. ML algorithms automatically identify the vital differences in abnormal data existing within standard data with high accuracy. Moreover, ML algorithms have strong generalization potentials; therefore, they can detect unknown attacks [170]. This section of the survey focuses on some of the standard ML algorithms that are frequently used in NIDS, the metrics, and benchmark datasets. ML-based NIDS is designed to monitor the host and its environment, analyze the systems’ behavior, generate alerts, and respond to all suspected attacks [171]. We consider ML methods under two main headings: supervised and unsupervised learning methods. There are other schools of thought in which there are more modern types classified as semi-supervised learning methods.

Supervised learning methods require classified and labeled data. In contrast, unsupervised learning extracts valuable information from an unlabeled dataset. To implement ML for any activity, three main important facets—time, space, and output—must be considered.

The development of anomaly detection systems using ML tools creates a generalization platform to generate futuristic normal or anomalous data to develop detection models. This generalization approach consists of generative and discriminative methods. Generative approaches construct ML-based models based on standard training data samples and perform several tests to determine how well they fit such a model [172]. Discriminative methods learn the difference between normal or anomalous data instances to create a model based on the results [173]. NIDS implemented in IoTs can be categorized based on the type of traffic and network structure. The detection is classified based on packet analysis, packet parsing, and payload analysis. Table 5 provides a summary of some of the machine learning-based NIDS and their metrics. In [174], the authors used recurrent neural network (RNN) deep learning to detect the intrusion. Their research focused on the analysis of ARM-Based IoT devices against malware with 98 % efficiency using 10-fold cross-validation consisting of two-layer neurons. As described in Saeed et al. [175], the energy-efficiency IoT devices operating with a batteries source cannot hold highly computationally intensive NIDS algorithms. Hence, choosing the right ML to create an intelligent security model for IoT systems is critical. The researcher must include critical evaluation processes such as the resources, power consumption, and performance of the IoT system. There are numerous ML-based NIDS for IoT in the research domain with different techniques. Azmoodeh et al. [176] used a deep Eigenspace learning approach to detect malware in IoT devices. Their proposed NIDS identifies malware that is located in the device’s Operational Code (OpCode) sequence. They generated the dataset for this experiment, making it difficult to benchmark their results with existing systems. In [177,178], the authors designed an intrusion detection system using a one-class support vector machine (SVM). Their one-class SVM is effective in providing good analysis with well-behaved feature vectors. However, the authors used deep belief networks (DBNs) for robust feature engineering. They show comparable anomaly detection while reducing the training and testing time. La et al. [179] modeled a Bayesian game theory-based NIDS to detect intrusion in the IoT system. From their analysis, attackers try to deceive security systems deployed in a network to initiate successful attacks against connected IoT devices in a network. Therefore, NIDS systems can use the honeypots technique to identify such intrusions. In [180,181], the authors developed an intrusion detection system that identifies an anomaly in IoT systems. The goal was to make an NIDS adaptive to its deployed environment and to detect intrusion based on the behavior of the attack. In [182], the authors used K-NN and Naive Bayes to identify intrusion. They tested their model using the NSL-KDD dataset. In [183,184], the authors proposed an NIDS model for detecting intrusion (BOTNET) for IoT devices using logistic regression. ML methods such as Decision Tree [185], K-Mean [186], DNN (using the NSL-KDD dataset) [187], and CNN (using the NGIDS-DS and ADFA-LD dataset for benchmark) [188] have been used by researchers to design NIDS for IoT systems. RNN, LSTM, GRU, and GAN [151,174] are also one of the most common neural network ML-based methods used by the research community to create NIDS for IoT. All the above models provide promising results but lack the analysis of the effects they pose on resources of the IoT devices.

### 7.1. Evaluation Metrics for NIDS in IoT

This subsection investigates the various metrics that are mostly used and also other metrics that are not widely considered in NIDS design and implementation for IoTs. The fundamental purposes of evaluating NIDS are:Compare two or more NIDS on a similar scale.Evaluate the required performances.Determine the best configuration of the NIDS.

#### 7.1.1. Performance and Evaluation Metrics Used in ML-Based NIDS for IoT Systems

Cardenas et al. [192] proposed that the Bayesian detection rates, the cost effects of failure detection rates [193], and the detection capabilities of any NIDS must be critical metrics to study. In [194], NIDS were evaluated based on the scorecard metrics in real-time and distributed systems. They argued that the data payload of the packets under analysis by an NIDS must contain realistic content. From their conclusion, the best way to evaluate any NIDS is by using real-time data from the deployed devices’ network rather than modeling or simulation. Data packets classified as an attack in a network using some ML-based NIDS algorithm may be regarded as legitimate packets by other classification models. Researchers use many metrics to evaluate NIDS. No single metric seems adequate to measure the capability and performance of NIDS. According to [195], the performance of an NIDS can be determined based on the Detection Rate (DR) and the False Alarm Rate (FAR). All the above analyses indicate that the performance of the NIDS designed for an IoT use case may differ. The implementation process, type of attack, IoT device’s resource, and the ML used during the NIDS contribute to the type of metrics.

However, the most common metrics used by researchers include Accuracy, Recall/Sensitivity, and F1-Score. Metrics such as false negative rate (FNR) (the number of negative samples predicted expressed as a ratio to the available, total positive samples) and false positive rate (FPR) (the ratio of positive samples predicted to the total predicted positive samples) are used to measure the performance of an NIDS for IoT systems.

#### 7.1.2. Other Required Metrics

When designing NIDS for massive IoT devices, researchers must consider some critical issues. Conventional IoT devices are energy resource-constrained. It is hard to recharge or replace the batteries, especially in a dynamic-isolated environment [196]. Some IoT devices are also deployed for specific jobs and therefore equipped with limited computational power and limited storage capacities. Hence, the power consumption and the IoT devices’ storage capacities are both critical factors in designing NIDS.

For instance, an IoT device equipped with low operating rate and limited storage capacity will require a constant power supply in order to measure the environment condition in a remote area. The IoT device has to transmit the measured weather data in a timely manner to a proximal MEC platform for processing [197]. Therefore, any NIDS design for such a system must be evaluated with these hardware settings. Most current research works ignored the importance of accommodating these critical considerations in their designs and performance evaluations.

### 7.2. Datasets Used in IoT Systems NIDS Design

The dataset is an essential part of an ML-based NIDS design. It consists of features observed from normal and abnormal operations of the targeted systems. For IoT networks, innovative methods and detecting algorithms necessitated a well-designed dataset. The most common data generation sources for NIDS for IoT are network packet extraction flows, system logs, and sessions. Building a dataset specifically for IoT NIDS can be complicated and time consuming. Table 6 shows an overview of several selected public datasets widely used by the research community for NIDS design.

Most researchers choose to create their own datasets to train the ML-based NIDS. Regardless of the difficulty in dataset construction, a common and well-established benchmark dataset is required in order to evaluate and to compare models. Researchers have used several datasets to train NIDS. The most commonly used datasets in the literature are the KDD-Cup’99 [198], which was created for the KDD competition, and it contains 41 attributes similar to a NetFlow dataset.

The UNB-ISCX 2012 [199], CICIDS2017 [200], and the AWS(CSE-CIC-IDS2018) [201] datasets were created by the Canadian Institute of Cybersecurity. According to the authors [200], the dataset was based on five days of normal and attacks traffic data. The CICIDS2017 contains most of the necessary modern and updated attack criteria such as DoS, DDoS, Brute Force, XSS, SQL Injection, Infiltration, Portscan, and Botnet. The dataset has 80 features extracted using the flow meter [202]. Another popular dataset currently used as a benchmark for NIDS in IoT is the $UNSW-NB_{15}$, which was created by the defense force academy of University of New South Wales Australia [203,204]. The $UNSW-NB_{15}$ was created based on the current attack categories through realistic network operations. The $UNSW-NB_{15}$ dataset consists of ten (10) categories of attacks: Analysis, Backdoor, DoS, Exploit, Fizzers, Generic, Reconnaissance, Shellcode, and Worm. Recently, there are several open source datasets that target different IoT attacks. In [205], the authors simulated a variety of network attacks in an IoT environment. Their dataset consists of benign, mitm-arpspoofing, DoS-synflooding, scan-hostport, scan-portos, and Mirai Botnet. Ullah et al. [206] created a dataset for IoT NIDS known as IoTID20. The authors proposed the IoT botnet dataset, which is suitable for DoS attacks. Moustafa [207] proposed a new dataset called TON_IoT, which contains IoT/IIoT service telemetry, as well as Operating System logs and IoT network traffic, obtained from a realistic approximation of a medium-scale network at the UNSW Canberra Cyber Range and IoT Labs.

The OPCUA dataset [208] was created by developing and injecting several attacks on a CPPS testbed based on OPC UA to allow users to assess the efficiency of various strategies for built Intrusion Detection Systems (IDS) in the industrial environment. Table 6 provides detailed information about NIDS in IoT systems. Datasets such as the ELEGANT dataset [209,210,211] targets DoS/DDoS attacks in IoT and SDN-based IoT networks, [212] detects anomalies in industrial IoT, Mqtt-iot-ids2020 dataset [213] detects network intrusion in MQTT protocol IoT network, Edge-IIoTset dataset [214] was also created for industrial IoT to detect DoS/DDoS attacks, Information gathering, Man in the middle attacks, Injection attacks, and Malware attacks. The X-IIoTID dataset [215] is a dataset for fitting the heterogeneity and interoperability of industrial IoT systems that is network and device agnostic. The final dataset on the NIDS IoT system radar is the IoT-BDA Botnet Analysis Dataset [216], which was created to make it easier to refine and create host and network-based IoT botnet detection systems. All the above datasets are publicly available for researchers to design robust NIDS to secure the IoT system.

**Table 6 sensors-22-03744-t006:** Some Popular Public Datasets for NIDS for IoT.

Dataset Description	Telemetry	Attack Types	Year Created
NSL-KDD [217]	General Purpose		2009
UNB-ISCX 2012 [199]	General Purpose	DoS, DDoS, Brute Force, XSS, SQL Injection, Infiltration, Portscan, and Botnet	2012
$UNSW-NB_{15}$ [203,204]	General Purpose	Analysis, Backdoor, DoS, Exploit, Fizzers, Generic, Reconnaissance, Shellcode, and Worm	2015
CICIDS2017 [200]	General Purpose	DoS, DDoS, Brute Force, XSS, SQL Injection, Infiltration, Portscan, and Botnet	2017
AWS(CSE-CIC-IDS2018) [201]	General Purpose	DoS, DDoS, Brute Force, XSS, SQL Injection, Infiltration, Portscan, and Botnet	2018
IoT Network Intrusion Dataset [205]	IoT	Benign, mitm-arpspoofing, DoS-synflooding, scan-hostport, scan-portos, mirai-udpflooding, mirai-ackflooding, mirai-httpflooding, mirai-hostbruteforce	2019
IoTID20 Dataset [206]	IoT	DoS/DDoS attacks	2020
TON_IoT Dataset [207]	IoT, IIoT, MEC, Fog	DoS, DDoS and ransomware	2019
OPCUA dataset [208]	IoT	DoS, Eavesdropping, Man-in-the-middle, Impersonation, Spoofing attacks	2020
IOT DOS AND DDOS ATTACK DATASET [209,210,211,218]	IoT	DoS and DDoS attacks	2021
IOT HEALTHCARE SECURITY DATASET [219]	IoT, SDN, IIoT	Normal and IoT attack traffic	2021
Mqtt-iot-ids2020 DATASET [213]	IoI, IIoT	Normal operation, aggressive scan, UDP scan, Sparta SSH brute-force, and MQTT brute-force attack.	2020
Edge-IIoTset DATASET [214]	IoT, IIoT, MEC	DoS/DDoS attacks, Information gathering, Man in the middle attacks, Injection attacks, and Malware attacks	2022
NSS MIRAI DATASET [220]	IoT	Mirai type botnet attack	2020
WUSTL-IIOT-2021 [221]	IoT	Command Injection, DoS, Reconnaissance, Backdoor	2021
IOT-BDA BOTNET ANALYSIS DATASET [216]	IoT	Port scanning, exploitation, C2 communications and DDoS	2021
THE BOT-IOT DATASET [222]	IoT	DDoS, DoS, OS and Service Scan, Keylogging and Data exfiltration attacks	2019
X-IIoTID dataset [215]	IoT, IIoT	Brute force attack, dictionary attack, and the malicious insider, reverse shell and Man-in-the-Middle	2021

Note: Network were captured as a Pcap. Users can extract features and records based on their preference.

## 8. Discussion and Future Trends

With an extensive increase in different IoT architectures ranging from well-defined documentation to unstructured standards, the IoT is confronted with new security issues. Due to the increase in cyber-attacks and the diversified capabilities of the IoT devices, manufacturers and developers need to adopt to a common standard when designing NIDS for these cyber-physical devices. The utilization of ML and the concept of MEC should be considered as a significant component in the NIDS design for IoT. This section examines a systematic process for choosing NIDS for IoTs and proposes an NIDS framework using MEC.

### 8.1. Choosing NIDS for IoT Systems

NIDS developers must consider some major factors when choosing the NIDSs for IoT systems. Firstly, the available resources of the IoT system must be analyzed, especially when the edge or fog computing platform can be utilized. The next factor to examine is how to implement a proper NIDS. Figure 4 summarizes the process of choosing an NIDS for the IoT systems. The specific IoT application also should be considered to choose the type of NIDS. Assuming a network consists of several IoT systems connecting to an MEC platform, a distributed anomaly-based ML model will be adequate.

If a single IoT device is installed remotely with limited resources, a good choice of NIDS for such a system will be a signature-based with centralized placement. Such a system should be updated frequently to identify new attacks. A hybrid-based NIDS will work well in an IoT MEC environment with equipped resources. In Figure 4, a link joining the anomaly-based and the ML-based NIDS demonstrates that combining the two approaches will be a suitable approach. The single arrow pointing to a method indicates that such a method alone will be a suitable choice. It is important to consider all the above scenarios in order for the manufacturers and developers to select the right NIDS for a certain IoT system.

### 8.2. Proposed NIDS Model for IoT Systems

A proposed model of distributed NIDS comprising of the IoT end-dive, edge, and the cloud has been elaborated in Figure 5. As explained in previous sections, IoT MEC has emerged to help resolve the resource constraints in IoT. Therefore, designing NIDS utilizing MEC will allow the implementation of a sophisticated high-end security system for IoT devices. Moreover, using the proposed model will help to control the amount of traffic that needs to be analyzed by the NIDS model on the MEC platform. The model does not require extensive resources on the edge platform to control the wide-range connected IoT devices.

In our proposed model, the training and retraining are located in the cloud, since it contains enough resources. Two different models, including a highly in-depth NIDS and a lightweight binary NIDS, can be created using the same dataset in the cloud. The lightweight model will be implemented in the IoT devices. The lightweight NIDS performs binary classification to check the availability of intrusion in the IoT end device. If an intrusion is detected, it then forwards the captured data to the edge platform for in-depth intrusion analysis. The results of the two models are compared and the decision is made. This will help reduce the false positive rates, which is a significant problem in the ML-based NIDS. Retraining is scheduled when there are new intrusions detected. Since MEC platforms can monitor and provide resources for a large number of IoT devices, the lightweight model will detect and control whether in-depth analysis should be performed on edge.

## 9. Conclusions

The necessity to secure IoT systems has stimulated many novel solutions to NIDS design. This paper has conducted a comprehensive review on NIDS that leverages MEC and ML. A taxonomy and tabular classification of detection methods, NIDS placement strategies, security threats, and validation methods were illustrated. We have discovered that there is a large body of theoretical studies. However, real-world NIDS development has not been fully testified yet. There are no specific detection techniques and deployment methods that have been accepted as standards to secure IoT systems. It still requires more effort to design a practical NIDS solution that effectively detects cyber-attacks in real IoT systems. Furthermore, critical IoT system evaluation metrics such as energy consumption, processing, and storage efficiency are not considered in most of the related studies.

Through our thoughtful review, we have identified several potential research directions. Firstly, future NIDS for IoT systems should focus on addressing the following issues: (1) improve the effectiveness of NIDS; and (2) demonstrate how their proposed system can be implemented in realistic IoT-MEC infrastructure. Secondly, researchers need to consider the critical metrics discussed in Section 7 for future NIDS design. We are confident that this survey and our proposed framework will serve as references and guidelines for researchers developing NIDS for IoT systems.

## Figures and Tables

**Figure 1 sensors-22-03744-f001:**
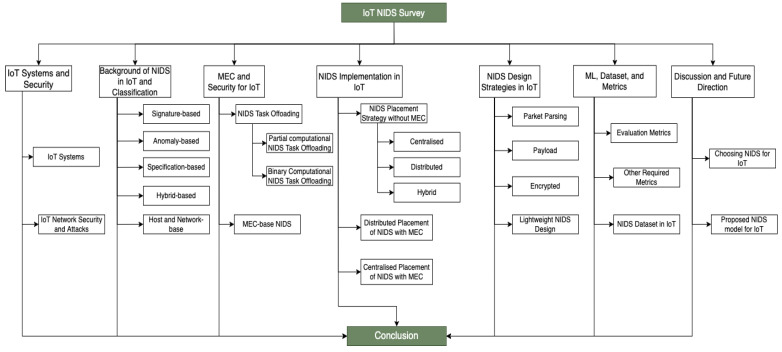
Taxonomy of Survey.

**Figure 2 sensors-22-03744-f002:**
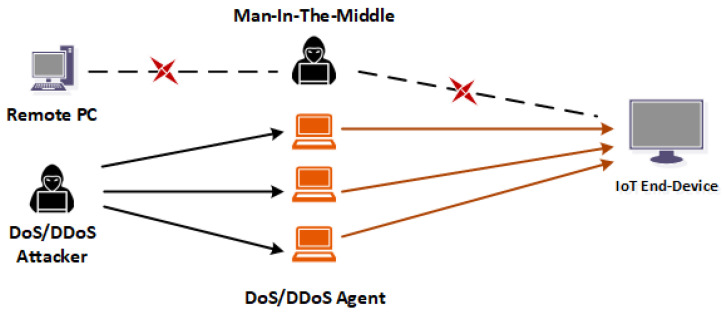
Network Attack.

**Figure 3 sensors-22-03744-f003:**
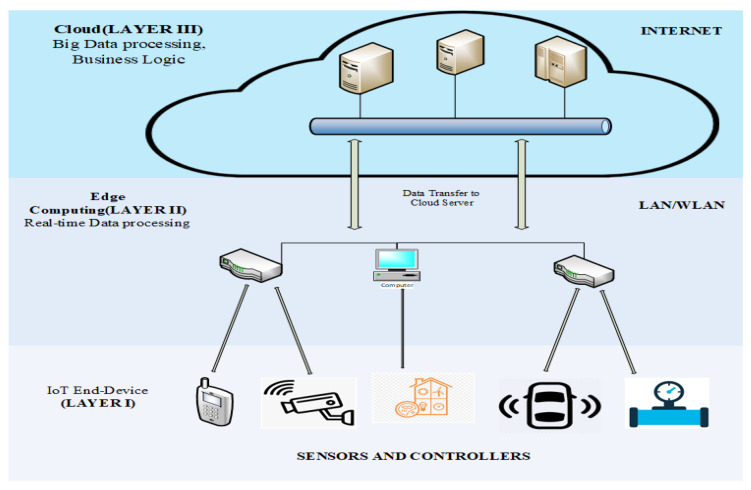
MEC Architecture.

**Figure 4 sensors-22-03744-f004:**
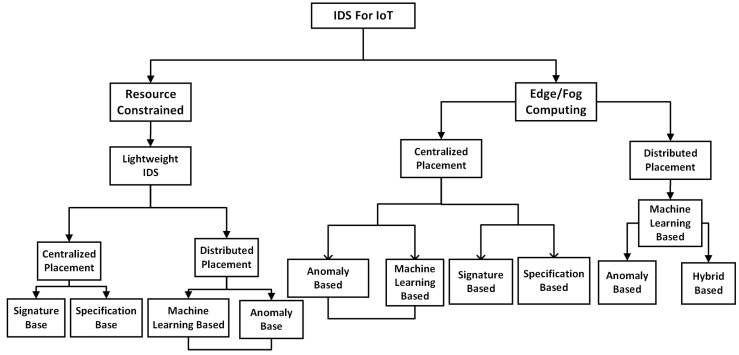
Structure of Choosing NIDS for IoT.

**Figure 5 sensors-22-03744-f005:**
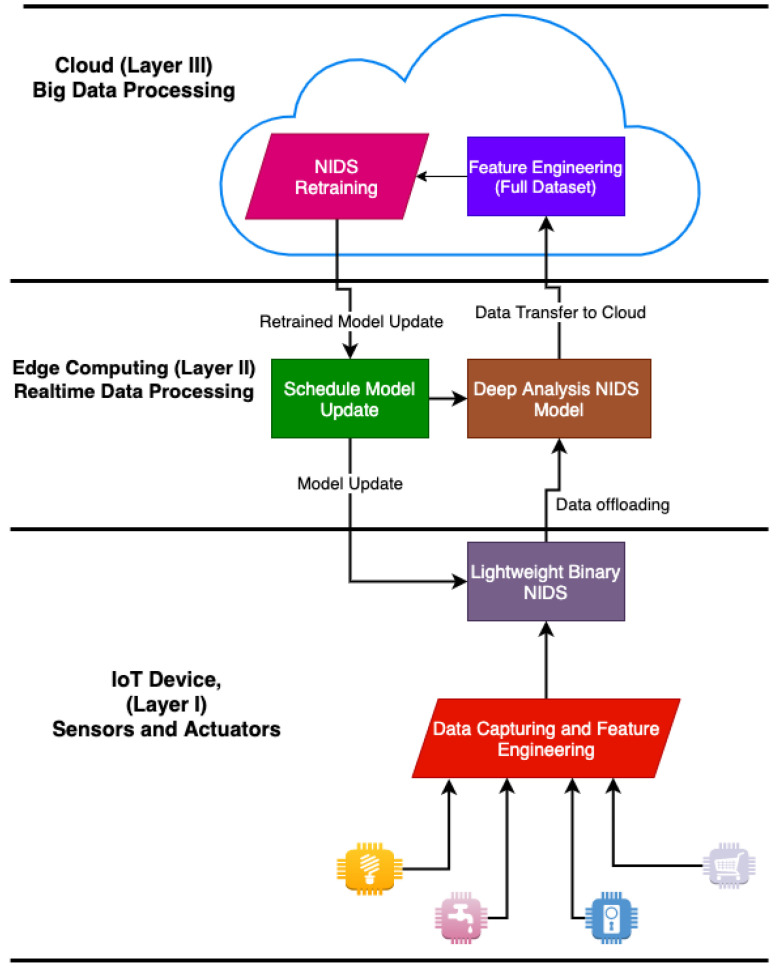
Proposed NIDS Model.

**Table 1 sensors-22-03744-t001:** Summary of Some Important Acronyms.

Acronym	Short Definition
IoT	Internet of Things
NIDS	Network Intrusion Detection System
ML	Machine Learning
NIDS	Intrusion Detection System
MEMS	Micro-electromechanical System
OT	Operational Technology
VHD	Virtual Honeypot Device
IEEE	Electrical and Electronics Engineers
6LoWPAN	IPv6 Low-Power Wireless Personal Area Network
LPWAN	Low Power Wide Area Network
DoS	Denial-of-Service
DDoS	Distributed Denial-of-Service
MITM	Man-In-The-Middle
WSN	Wireless Sensor Network
RPL	Routing Protocol for Low Power and Lossy Network
NFV	Network Functions Virtualisation
ICN	Information-Centric Network
SDN	Software-Defined Network
MEC	Mobile Edge Computing
VM	Virtual Machine
RL	Reinforcement Learning
LA	Learning Automata
MLP	Multilayer Perceptron
R2L	Remote-to-Local
U2R	User-to-Route
CNN	Convolutional Neural Network
DIDS	Distributed Denial of Service
LSTM	Long Short-Term Memory
RNN	Recurrent Neural Network
AIS	Artificial Immune System
PCA	Principal Component Analysis
QoS	Quality of Service

**Table 2 sensors-22-03744-t002:** Common Attacks in IoT-MEC.

Source	Attacks	Mode of Attack Initiation
[38,39,40]	Spoofing	Impersonation
[41,42,43]	Denial of Service (DoS)	Network Flooding
[44,45,46]	Distributed Denial-of-Service (DDoS)	Network Flooding
[47,48]	Jamming	Fake Signaling
[49,50]	Man-In-the-Middle	Eavesdropping Packets
[51,52]	Privacy Leakage	Attack Authentication Storage
[53,54]	Marai Botnet Attack	Malware Implant on Devices
[55,56]	Sybil Attack	Creates Anonymous Identities
[57,58]	AI-Based Attacks	Creates AI-powered Tools

**Table 3 sensors-22-03744-t003:** Different between Host-Based and Network-Based IDS.

	Host-Based NIDS	Network-Based IDS
Data Source	System call logs	Captured network traffic
Placement Strategy	Locally on the hosted device or machine	Specific IoT devices on the same subnet
Detection Rate	Low, difficult to detect new attacks	High, can detect new attacks in real time
Threats Trace-ability	Based on system calls	Trace intrusion based on network addresses and timestamps
Limitations	Cannot analyse network attacks, rules created can be obsolete, depends on the operating system (OS)	Monitor only network traffic within a specific subnet

**Table 5 sensors-22-03744-t005:** Analysis of ML-Based NIDS in IoT.

Reference	ML Methods	Precision	Recall	Accuracy
Mayhew et al. [189]	Behavior-Based Access Control	99.1%	-	-
Haddadpajouh et al. [174]	RNN, LSTM	-	-	98.18%
Saeed et al [175]	RNN	-	-	97.23%
Azmoodeh et. al. [176]	Deep Eigenspace learning	98.59%	98.37%	99.68%
Erfani et al [177]	One-Class SVM	-	-	-
La et al. [179]	Bayesian Game Theory	-	-	-
Arrington et al. [181]	Behavioral Modeling	-	-	-
Li et al. [190]	KNN	98.5%	-	-
Pajouh et al. [182]	Naïve Bayes	84.86%	-	-
Ghosh and Mitra. [183]	Logistic Regression	-	-	93.26%
Prokofiev et al. [184]	Logistic Regression	94.0%	98.0%	97.30%
Singh and Neetesh [191]	Self-Organizing Map	64%	-	-

The “Applicable with Edge” column was established based on our observation on various system implementation.

## Data Availability

Not Applicable.

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
