# Peer review of "Intrusion Detection in Internet of Things Systems: A Review on Design Approaches Leveraging Multi-Access Edge Computing, Machine Learning, and Datasets"

_sensors, 2022, doi:10.3390/s22103744_

Round 1
Reviewer 1 Report
See attached file.

Author Response
We appreciate your immense comments on our work. We have made all the required changes you suggested.
- Comment: Section 1.2 already explains the motivations and contributions of this survey. However, I suggest the Authors to better explain in this Section how their survey enriches the big picture of the surveys dealing with these topics;
Response: Thank you for this comment. We have added an additional content to subsection 1.2 mark in red in the main document. we have included "Moreover, this paper add to knowledge the comprehensive reviews of state-of-the-art design of NIDS for the resource-constraint IoT using the MEC, the implementation strategies, and the IoT dataset used. The study extends the design approaches used by researchers and how the proposed methods fit into NIDS design for IoT systems. We also proposed an NIDS framework for the IoT utilising MEC architecture and demonstrated the possible ways of choosing NIDS for the IoT devies base on some conditions." to enrich the content.
Reviewer 2 Report
This manuscript introduces a review of state-of-the-art network intrusion detection systems (NIDS) and security practices for IoT networks. The authors claims that they perform a comparative analysis on the public available datasets, evaluation metrics, and deployment strategies employed in the NIDS design. The reviewer do not found a solid technical information and useful metrics comparisons that may recommend a research path for the readers. The reviewer recommends rejecting this manuscript.
Author Response
We appreciate your constructive review of our paper and the comments you provided. However, the main aim of this review is to provide the IoT community with research directions in network intrusion detection system (NIDS) design to secure the IoT system.
As outlined in the manuscript, IoT devices have resource constraints. Hence, conventional NIDS cannot be implemented on them. There are new paradigms of NIDS design strategies employed by the research community to overcome this problem. Multi-access mobile edge computing (MEC) is one of the modern technologies that mitigate the resource-constraints in IoT devices. Researchers are using this technology to design security systems to protect IoT devices. Another major problem with NIDS design for the IoT system is the source of dataset for evaluation. Most of the research work that targets this domain uses old datasets that are not IoT centric.
Hence, this paper provides a comprehensive review of the state-of-the-art NIDS strategies, focusing on MEC, placement strategies, implementation strategies, and some of the datasets that focus on attacks created with IoT systems. We also climaxed our review with our proposed NIDS framework that will aid researchers who wish to design NIDS for IoT systems utilizing MEC. The review as a whole also provides information for continuing learning in IoT security. For example, section 6 examines the NIDS design strategies that are suitable for the IoT system.
We have also updated the paper by proofreading the content and correcting all grammatical errors found. We hope the above explanation will provide you with a brief insight into our paper and help with the review process.
Reviewer 3 Report
This paper studies the intrusion detection issue for IoT devices with the focus on Multi-access Edge Computing using machine learning. The scope of the paper is very interesting and well-established. I would suggest authors to create a sub-section that describes the general structure of IoT with all parts (device layer, edge layer, fog layer and cloud). This is important to give a brief information about IoT era. Also, I did not see any surveyed work with reinforcement learning, please check it.
Author Response
Your detailed feedback on our work is much appreciated. We have, however, implemented all of the necessary improvements indicated in your review. Comment: I would suggest authors to create a sub-section that describes the general structure of IoT with all parts (device layer, edge layer, fog layer and cloud). This is important to give brief information about IoT era. Also, I did not see any surveyed work with reinforcement learning, please check it
Response: Thank you for this comment. We have created a new section called "IoT-MEC Architecture" in section 2.2, which provides a brief description of the IoT device layer, MEC/Fog layer, and the cloud.
We have also included reviews of NIDS created using reinforcement learning in sections 3.1 and 3.2. The references include [68, 69] in section 3.1 and [79,80] in section 3.2. The details of the above responses have been marked red in the manuscript.
Reviewer 4 Report
The authors have presented an in-depth analysis of intrusion detection in IoT systems. This review can present valuable insights to the upcoming researchers.
Author Response
Thank you for spending time reviewing our paper. We appreciate your review comments.
Round 2
Reviewer 1 Report
The paper notably improved, and I believe the Authors did a great job. I am only left with two minor remarks, which have not to be considered as mandatory.
- Please, avoid the use of genitives;
- In Section 2.2.2 consider citing 10.1109/BigData.2018.8622117 10.1109/CAIS.2019.8769500 and https://doi.org/10.3390/app12031497 because they propose interesting edge/fog IoT architectures.
Once again, well done!
Reviewer 2 Report
accept